

# Rethinking the deployment of static chambers for CO2 flux measurement in dry desert soils

Nadav Bekin, Nurit Agam

[1]Blaustein Institutes for Desert Research, Ben-Gurion University of the Negev, Sede-Boqer Campus, 84990, Israel

*Correspondence to:* Nurit Agam (agam@bgu.ac.il)

**Abstract.** The mechanisms underlying the soil $CO_2$ flux (Fs) in dry desert soils are not fully understood. To better understand these processes, we must accurately estimate these small fluxes. The most commonly used method, static chambers, inherently alter the conditions that affect the flux and may introduce errors of the same order of magnitude as the flux itself. Regional and global assessments of annual soil respiration rates are based on extrapolating point measurements conducted with flux chambers. Yet, studies conducted in desert ecosystems rarely discuss potential errors associated with using static chambers in dry and bare soils. We hypothesized that a main source of error is the collar protrusion above the soil surface. During the 2021 dry season, we deployed four automated chambers on collars with different configurations in the Negev Desert, Israel. Fs exhibited a repetitive diel cycle of nocturnal uptake and daytime efflux. $CO_2$ uptake measured over the conventionally protruding collars was significantly lower than over the collars flushed with the soil surface. Using thermal imaging, we proved that the protruding collar walls distorted the ambient heating and cooling regime of the topsoil layer, increasing the mean surface temperatures. Higher soil temperatures during the night suppressed the flux driving forces, i.e., soil-atmosphere $CO_2$ and temperature gradients, ultimately leading to an underestimation of up to 50% of the actual Fs. Accordingly, the total daily $CO_2$ uptake by the soil in the conventionally deployed collars was underestimated by 35%. This suggests that desert soils are a larger carbon sink than previously reported and that drylands, which cover approximately 40% of Earth's terrestrial surface, may play a significant role in the global carbon balance.



## 1    Introduction

Soil respiration, i.e., the carbon dioxide ($CO_2$) efflux from the soil to the atmosphere, is among the largest components of the carbon balance in terrestrial ecosystems, contributing approximately 60 PgC to the atmosphere every year (Houghton, 2007). In arid and semi-arid environments, soil respiration is mostly considered to be restricted to short pulses of increased moisture availability from rainfall events, during which microbial metabolic activity increase rapidly, followed by long periods of desiccation and low to negligible soil respiration rates (Austin et al., 2004; Cable et al., 2008). In the last two decades, studies carried out in several deserts have challenged this paradigm, reporting a diel course of $CO_2$ exchange during dry periods, consisting of nocturnal $CO_2$ uptake and daytime efflux (Sagi et al., 2021; Lopez-Canfin et al., 2022). Researchers usually attribute this diel cycle to changes in soil temperatures and soil air pressure that leads to cycles of expansion/contraction of soil air, following the ideal gas law (Yang et al., 2020). These cycles change the surface $CO_2$ concentration and may generate a soil-atmosphere pressure gradient (Ganot et al., 2014), both driving forces for soil $CO_2$ flux ($F_s$). Another explanation is based on Henry's Law. It states that diurnal fluctuations of soil temperatures change the solubility of soil $CO_2$ in water films, which changes the concentration of gaseous $CO_2$ in soil pores, leading to the exchange of $CO_2$ between the soil and the atmosphere by diffusion (Fa et al., 2016). In saline/alkaline soils, this process is thought to cause a diel cycle of calcium carbonate ($CaCO_3$) precipitation/dissolution, which enhances $F_s$ (Hamerlynck et al., 2013; Fa et al., 2016). Yet, the factors controlling $F_s$ in dry desert soils and the partitioning between them are still under debate.

Furthermore, the ability to accurately estimate the soil $CO_2$ flux in desert soils at the very dry-end is controversial due to the potential for measurement-induced modifications to soil and atmospheric conditions that can introduce errors of the same order of magnitude as the flux being measured. This problem is exacerbated when using static chambers to measure flux, as the chambers inherently alter the conditions that affect the flux (Pumpanen et al., 2010 ; Parkin et al., 2012). During efflux, $CO_2$ concentration in the chamber builds up, decreasing the diffusion gradient between $CO_2$ in the soil pores and the chamber headspace, thereby altering $CO_2$ concentration within the top soil layer and reducing the flux (Pumpanen et al., 2004). Artificial changes in air pressure within the chamber headspace compared to the ambient atmosphere are another source of error (Bain et al., 2005; Lund et al., 1999).

There are additional sources of errors associated with the chamber-soil contact method (Ngao et al., 2006; Baram et al., 2022). Flux chambers are typically deployed on a collar (i.e., PVC pipe) that is inserted into the soil, with the upper 3-5 cm of the collar protruding above the soil surface to allow for chamber deployment. This practice modifies the soil surface temperature by shading a portion of the measured surface area. The non-representative soil surface temperature results in modified heat exchange between the soil and the atmosphere, as well as a modified soil temperature profile (Ninari and Berliner, 2002). Soil microbial and physical processes that drive $F_s$ are susceptible to changes in soil temperature  (Cable et al., 2011), and thus shading the soil surface can lead to errors in Fs measurements. While these effects are likely minimal in temperate, vegetated areas, they could be significant in dry bare soil, partly because fluctuations in surface temperatures are not regulated by vegetation cover as in humid environments. Desert soils also have lower specific heat capacity than soils in humid environments due to lower water content (Hillel, 1998). The lower water content also means that a larger portion of the available energy is invested in soil heating rather than stored as latent heat during evaporation (Brutsaert, 1982). However, studies using static chambers in desert ecosystems rarely discuss potential errors associated with



the unique characteristics of desert soils. Moreover, to our knowledge, the effect of collar height above the surface
on soil surface temperature and, consequently, on $F_s$ was never studied.
Under dry soil conditions, the depth to which the collar is inserted can also significantly influence the flux
measurements. The ideal insertion depth is debatable, as both shallow and deep collar insertion depths can lead to
errors, depending on climate and soil conditions. Inserting the collar to a  shallower depth than the depth to which
feedback from the chamber still affects gas concentrations may result in lateral diffusion, leading to
underestimation of the vertical flux (Healy et al., 1996). However, insertion depth of only 2.5 cm and a
measurement period of 10 minutes will reduce this underestimation to 1% for a soil with air-filled porosity of 0.3
$m^3\,m^{-3}$ (Hutchinson and Livingston, 2001). Hence, for short measurement periods (common today) and soils with
low effective diffusivity, errors resulting from lateral diffusion may be insignificant. With current static chamber
systems, even small $F_s$ measured in dry desert soils can be accurately quantified with much shorter measurement
periods of only 1-2 minutes (Yang et al., 2022), thus overcoming a significant drawback of the shallow collars.
Deep collar insertion, on the other hand, can lead to either overestimation or underestimation of the flux by
generating vertical mass flow of air along the collar walls or by facilitating root cutting, respectively (Heinemeyer
et al., 2011). Still, in most studies, collars are inserted to a depth of ~5-10 cm into the soil and, in some cases, to
a depth of 30-60 cm, while more than a third of all authors fail to report the collar insertion depth (Rochette and
Eriksen-Hamel, 2008; Cable et al., 2011; Fa et al., 2018; Jian et al., 2020; Sagi et al., 2021; Yang et al., 2022).
In this paper, we aimed to investigate the effect of collar height above the soil surface and collar depth of insertion
on $F_s$ in a dry bare desert soil. Given the small fluxes in these conditions, and the fact that regional and global
assessments of annual soil respiration are based on extrapolating point measurements conducted with flux
chambers (Jian et al., 2020), minimizing measurement errors associated with the collar deployment technic is
critical. Arid and semi-arid regions, which comprise approximately 40% of earth's terrestrial surface, constitute
the largest uncertainty on mean annual soil respiration estimations (Stell et al., 2021). Improving the accuracy of
$F_s$ measurements in desert environments is essential for enhancing our understanding of the terrestrial carbon
balance and our ability to predict climate change.
## 2    Materials and Methods
### 2.1    Research site
The study was carried out at the Wadi Mashash Experimental farm in the Northern Negev, Israel (31°04'14''N,
34°51'62''E; 360 m.a.s.l; 65 km SE of the Mediterranean Sea). The climate in the research site is arid, with an
average annual rainfall of 116 mm (IMS, 2021), occurring between October and April. The daily mean maximum
and minimum temperatures for January (winter) are 15.9 C° and 8.0 C°, respectively, while those for August
(summer) are 33.3 C° and 20.7 C°. During the summer season, the prevailing wind direction is NW due to the sea
breeze carrying water vapor from the Mediterranean Sea inland. The sea breeze reaches its peak at a wind speed
of 7 $m\,s^{-1}$ (at 10 m height) in the afternoon. The research is located on a largely bare plain of sandy-loam loess
soil (72.5% sand, 15% silt and 12.5% clay), partly covered by a biological soil crust over a thin physical crust,
with dry annual grasses and Shrubs.



### 2.2 Meteorological measurements

Air temperature and relative humidity (100K6A1A, BetaTherm, USA) were monitored along with wind speed
and direction as part of an eddy-covariance system (IRGASON, Campbell Scientific Inc.). Air temperature was
measured at 5-second intervals and averaged over 15-minute periods. Wind speed and direction were determined
from high-frequency measurements of 3D wind speed taken at 20 Hz intervals, then averaged over 30-minute
periods and stored in a data logger (CR6, Campbell Scientific Inc.). Net radiation was measured at a height of 2.4
m using a 4-component net radiometer (SN-500-SS, Apogee instrument Inc, USA) at 10-second intervals,
averaged over 15-minute periods, and stored in a data logger (CR5000, Campbell Scientific Inc.).

### 2.3 Soil $CO_2$ flux measurements

We measured $F_s$ using a non-dispersive Infrared Gas Analyzer with a range of 0-20,000 ppm and an accuracy of
1.5% of reading. The analyzer was connected to four automated non-steady-state chambers (LI 8100A- 104C, LI-
COR, Lincoln, USA). The chambers were closed on a pre-inserted collar every 30 minutes for a measurement
period of 60 seconds, with a 10-second dead band period to allow homogeneous air mixing within the system.
Each measurement started with a 90-second pre-purge and ended with a 45-second post-purge period.
We deployed the chambers on three types of collars (i.e., treatments): (1) The conventional type (CONV) - an 11
cm long collar, inserted 7.5 cm into the soil, leaving 3.5 cm of collar above the soil surface (Fig. 1); (2) The deep
type (DEEP) - an 11 cm long collar completely inserted into the soil, leaving the top of the collar flush with the
soil surface; and (3) The shallow type (SHAL) - a 2.5 cm long collar completely inserted into the soil, with the
top of the collar flush with the soil surface. Three collars from each type (1-3) were inserted into the soil two
months before measurements started. All collars had a inner diameter of 20 cm.

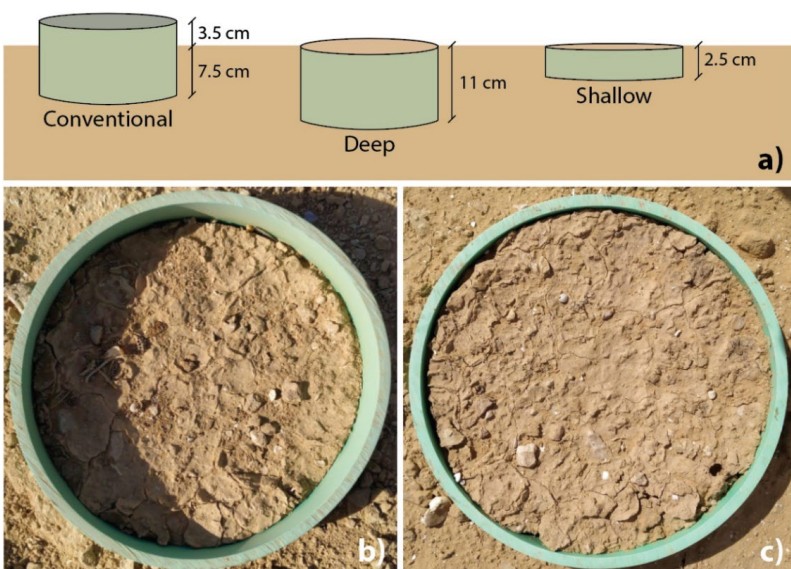

**Figure 1: a) The three types of collars used in this experiment. b) Photo of a conventional (CONV) collar. C) Photo of a collar flashed with the soil surface, representing the DEEP and SHAL treatments.**



We collected data between May and June of the 2021 dry season. Three chambers were rotated between the collars on a near-weekly basis (periods 1-6; Table 1), ensuring that each period consisted of at least five full and representative days. The fourth chamber was placed on an additional DEEP collar for the whole experiment duration (the permanent type - PERM). The chambers were rotated in two configurations (Table 1): during periods 1, 3 and 5, each chamber was set over a different treatment, e.g., in period 1, chambers were placed over collars CONV1, DEEP1, SHAL1; and during periods 2, 4 and 6, the three chambers were placed on the same treatment (SAME), e.g., in period 2, chambers were placed over collars CONV1, CONV2, CONV3.

**Table 1. Chamber placement during the 6 measurement periods - 12/05-29/06/2021**

| Period | 1 | 2 | 3 | 4 | 5 | 6 |
|---|---|---|---|---|---|---|
| Dates | 12-18/05 | 18-22/05 27-30/05 | 30/05-03/06 06-09/06 | 09-16/06 | 16-22/06 | 24-29/06 |
| Analyzed days | 12-16/05 | 19-21/05 28-29/05 | 31/05-02/06 07-08/06 | 09-14/06 | 17-21/06 | 25-29/06 |
| Treatment and replicate | CONV1 DEEP1 SHAL1 | CONV 1-3 | CONV2 DEEP2 SHAL2 | DEEP 1-3 | CONV3 DEEP3 SHAL3 | SHAL 1-3 |

One chamber (PERM) continuously measured soil $CO_2$ flux on the same collar throughout the experiment.

## 2.4 Ancillary soil measurements

The temperature profile in the soil was measured by self-made T-type thermocouples buried at depths of 0.5, 1, 2, 3, 4, 5, 10, 15, 20, 30 and 50 cm. The thermocouple buried at 0.5 cm provided a proxy for the soil surface temperature. The soil heat flux was derived using the combination method with three repetitions, using a soil heat flux plate (HFT3, Campbell Scientific Inc.) buried at a depth of 5 cm. Heat storage above the plates was derived from two self-made T-type thermocouples buried at depths of 1.25 and 3.75 cm, and soil water content was measured with a time-domain reflectometer (TDR-315, Acclima, Inc., USA) installed at a depth of 3 cm. The volumetric water content of the soil was lower than 3% throughout the experiment. Temperature profile and water content data were collected at 10-second intervals, and 15-minute averages were stored in a data logger (CR1000X, Campbell Scientific Inc.) and multiplexer (AM 16/32B, Campbell Scientific Inc.). Soil heat flux data were also collected at 10-second intervals, and 15-minute averages were stored in a data logger (CR5000, Campbell Scientific Inc.).

## 2.5 Radiometric surface temperature

A 24-hour field campaign was conducted on August 17-18, 2021. During the campaign, the surface radiometric temperature of the collars was acquired hourly using a thermal infrared camera (A655sc, FLIR, Wilsonville, USA), immediately before taking $F_s$ measurements.

## 2.6 Data analysis

To calculate $F_s$, a linear function was fitted to the change in $CO_2$ mole fraction over time for each measurement, using the software LI-COR SoilFluxPro 5.2.0 (LI-COR, Lincoln, USA). The fitting period, which usually lasted 20 seconds, started after air mixing within the chamber was achieved.



To decipher the differences between collars, and given the limited number of chambers, we derived an "average-
day" for each collar type (CONV, DEEP, and SHAL). First, five full representative days from each experiment
period (Table 1) were analyzed. Then, for each of the four chambers, an average diel course was calculated from
the 5 analyzed days, resulting in 4 average days per period. All average days from all periods (4 treatments × 6
periods= 24 average days) were then divided into 3 groups based on collar type (6 average days per treatment),
and a single average day per treatment was calculated as the mean of the 6 average days. Each time point in the
three treatment average days consists of 30 values (6 average days × 5 days per average).
The differences between the treatments were tested for significance using linear mixed models (LMMs), following
the approach developed by Spyroglou et al. (2021). We built a statistical model Using LMMs that predicted the
response variable (i.e., the mean daily cycle of $F_s$) as a function of treatment and time as fixed factors (fixed for
all data points), and each collar as a subject-specific factor (random effect). This allowed us to assess the effect
of treatment, but also the effect of time and individual collars on $F_s$, while incorporating all 24-hour time series
into a single model. Still, this model fails to defuse the autocorrelation between data points in each time series. To
address this, the LMM residuals were passed through an Autoregressive Integrated Moving Average (ARIMA)
model and then incorporated within the LMM as errors. The predicted $F_s$ values produced by the corrected model
were compared between treatments for each time interval separately using a two-tale t-test with a 95% confidence
interval. To avoid type I errors, the p-value was divided by the number of tests performed on each time point
according to the Bonferroni correction. Therefore, the corrected p-value used here is 0.05/6=0.008. The
differences between the treatments were also tested by comparing peak daily and daily accumulated efflux and
uptake value. This was executed using one-way ANOVA and a post hoc Tukey test with a 95% confidence
interval. The modeling process and statistical analysis were performed using "stats", "lme4" and "forecast"
packages in RStudio 4.1.1.
To analyze the collars surface temperature, the region of interest (ROI) for each thermal image was defined for
the collar's inner surface area using FLIR ResearchIR Max 4.40.35. The surface temperature of all pixels within
the ROI were then exported to RStudio to calculate statistical parameters used to compare treatments. The soil
surface emissivity was set to 0.95 for all images (Li et al., 2013).
**3   Results**
**3.1   Meteorological and soil conditions**
The experiment period was characterized by clear sky days, with similar diel patterns and magnitudes of incoming
short-wave and net-radiation (Fig. 2). Solar noon occurred at 11:30 every day of the experiment (UTC+02:00).
Sunrise and sunset occurred at 04:30-05:00 and 19:00, respectively. The daily minimum and maximum air and
soil surface temperatures were 19.45±2.3 and 34.5±2.7 C° (air) and 17.7±2 to 49.6±2.2 C° (soil surface),
respectively. The mean daily range was 13.7±1.0 and 31.8±1.2 C°, for the air and the soil surface respectively,
with a slight variation between the experiment weeks. The soil surface temperature regularly dropped below air
temperature at night (Fig. 2B). The prevailing wind direction was NW, peaking in the afternoon at a mean speed
of 6.2±0.2 m s⁻¹ (2 m height).



**Figure 2: Time series with half hourly data of environmental variables measured at the Wadi Mashash Experimental farm during the 2021 summer season. A) Incoming shortwave radiation and net radiation. B) Air and soil surface temperatures measured at 0.5 cm depth. C) Wind speed is color-coded according to wind direction: north (N), north-west (NW), west (W), south-west (SW), south (S), south-east (SE), east (E), and north-east (NE). D) The soil CO₂ flux measured by the permanent chamber (PHARM).**

Soil $CO_2$ flux measured on the permanent collar followed a consistent diurnal pattern throughout the experiment (Fig. 2d), confirming that the weekly periods can be used to test differences between treatments. Starting from the afternoon (mean time 13:30), negative $CO_2$ flux (i.e., uptake; from the atmosphere to the soil) occurred, peaking,



on average, at a flux of $-0.4\pm0.04$ µmol m$^{-2}$ s$^{-1}$ (at 18:30). Then in the early morning (06:00), the flux reversed,
and positive $CO_2$ flux (i.e., efflux; from the soil to the atmosphere) increased sharply until 08:30, when a daily
maximum of $0.71\pm0.08$ µmol m$^{-2}$ s$^{-1}$ was observed. After that, efflux gradually decreased until the afternoon.
**3.2   The effect of collar type on soil $CO_2$ flux**
The daily temporal dynamic of $F_s$ shows little variation among the different treatments. However, the rate of
increasing $CO_2$ efflux in the early morning, measured by the CONV collars, was lower than in the other treatments,
as evidenced by the curve's concave nature (Fig. 3). Consequently, the daily maximum $CO_2$ efflux of CONV
occurred at 08:30, an hour later than in the other treatments. The SHAL collars were also different from the other
treatments in the timing of $CO_2$ uptake onset, occurring each day between 12:00-12:30, two hours before uptake
started in the other treatments (Fig. 3).

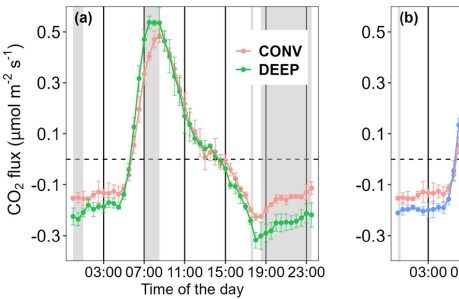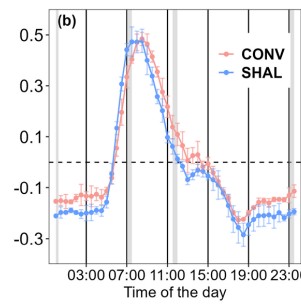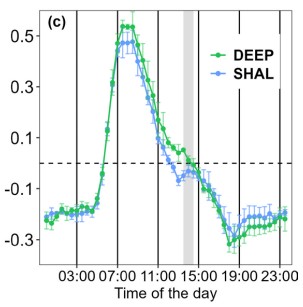


**Fig. 3. Mean daily cycles of the soil $CO_2$ flux measured in the following collar types- A) The conventional (CONV) and**
**deep (DEEP) insertion types. B) The conventional (CONV) and shallow (SHAL) types. C) The shallow (SHAL) and**
**deep (DEEP) types. Error bars denote two standard deviations (n=30). Gray areas represent periods in which**
**differences between the treatments were statistically significant (p-value<0.008).**
The LMM model, combined with time series analysis, yielded statistically significant results ($P<0.008$) for the
differences in $F_s$ between CONV and DEEP during the morning (07:00-08:30) and the evening/night (17:30-
01:00). In fact, $F_s$ of CONV were consistently lower than in DEEP. The relative differences peaked at 06:00 and
23:30, when mean daytime $CO_2$ efflux and nocturnal $CO_2$ uptake were 56 and 53% lower in the CONV than in
the DEEP. $F_s$ measured in the CONV collars were also significantly lower than SHALL, by a maximum of 41%,
but for shorter periods around noon and midnight. $F_s$ measured in the DEEP collars were only significantly
different from SHAL ($P<0.008$) from 13:30 to 14:30.
The mean peak daily efflux measured in the DEEP treatment differed significantly from the other two treatments
($p<0.05$), while no statistically significant difference in peak efflux was found for SHAL and CONV (one-way
ANOVA and Tukey post hoc test). The differences between the total daily amount of $CO_2$ emitted during the day
measured in SHAL and CONV were also insignificant ($p>0.05$; Table 2). In contrast, the total daily amounts of
$CO_2$ uptaken by the soil in the CONV collars were significantly lower than in the SHAL and the DEEP collars
(Table 2), which may lead to erroneous estimations of daily net $CO_2$ exchange.




**Table 2. Summary of main features- the mean daily cycles of *Fs***

| Period | Treatment | Max CO$_2$ efflux | Max CO$_2$ uptake | Total uptake | Total efflux |
|---|---|---|---|---|---|
| | | $\mu$mol m$^{-2}$ s$^{-1}$ | $\mu$mol m$^{-2}$ s$^{-1}$ | g m$^{-2}$ | g m$^{-2}$ |
| 1 | CONV1 | 0.51±0.08 | -0.28±0.04 | 0.43±0.077 | 0.29±0.04 |
| | DEEP1 | 0.61±0.06 | -0.38±0.05 | 0.54±0.05 | 0.39±0.06 |
| | SHAL1 | 0.59±0.06 | -0.38±0.06 | 0.57±0.10 | 0.32±0.05 |
| 2 | CONV1 | 0.47±0.04 | -0.26±0.03 | 0.30±0.05 | 0.33±0.04 |
| | CONV2 | 0.51±0.06 | -0.26±0.03 | 0.28±0.04 | 0.37±0.03 |
| | CONV3 | 0.52±0.07 | -0.27±0.02 | 0.31±0.06 | 0.32±0.04 |
| 3 | CONV2 | 0.57±0.09 | -0.25±0.04 | 0.36±0.11 | 0.39±0.04 |
| | DEEP2 | 0.61±0.07 | -0.36±0.03 | 0.53±0.13 | 0.34±0.09 |
| | SHAL2 | 0.58±0.10 | -0.35±0.03 | 0.49±0.12 | 0.33±0.08 |
| 4 | DEEP1 | 0.64±0.08 | -0.38±0.04 | 0.47±0.11 | 0.41±0.05 |
| | DEEP2 | 0.67±0.11 | -0.40±0.03 | 0.52±0.14 | 0.40±0.05 |
| | DEEP3 | 0.57±0.10 | -0.34±0.07 | 0.43±0.14 | 0.33±0.05 |
| 5 | CONV3 | 0.55±0.04 | -0.27±0.03 | 0.41±0.04 | 0.33±0.03 |
| | DEEP3 | 0.60±0.01 | -0.30±0.02 | 0.47±0.03 | 0.36±0.04 |
| | SHAL3 | 0.48±0.04 | -0.28±0.03 | 0.44±0.02 | 0.28±0.04 |
| 6 | SHAL1 | 0.56±0.03 | -0.32±0.01 | 0.46±0.11 | 0.34±0.01 |
| | SHAL2 | 0.52±0.04 | -0.28±0.03 | 0.37±0.08 | 0.28±0.03 |
| | SHAL3 | 0.48±0.02 | -0.27±0.02 | 0.35±0.09 | 0.29±0.03 |

Each value in the table is an average of 5 days ± one standard deviation.

### 3.3    The effect of collar type on the radiometric soil surface temperature

The mean and range of soil radiometric surface temperatures in the CONV collars were higher than in the DEEP
and SHAL collars, even at midday (Fig. 4). At 16:00, the three treatments all exhibited a mean surface temperature
of 40 °C, but the range of surface temperatures in the CONV collars doubled those of the other treatments. During
the night, the mean surface temperature of the CONV collars was 0.5-1 °C higher than in the DEEP collars and
0.5-0.9 °C higher than in the SHAL collars. After sunrise, the surface temperatures of the CONV and SHAL
increased faster than in the CONV collars up to 07:00. Later, the mean surface temperature of DEEP and SHAL
maintained a similar distribution over time, while the range and mean surface temperature in the CONV increased
sharply (Fig. 5).



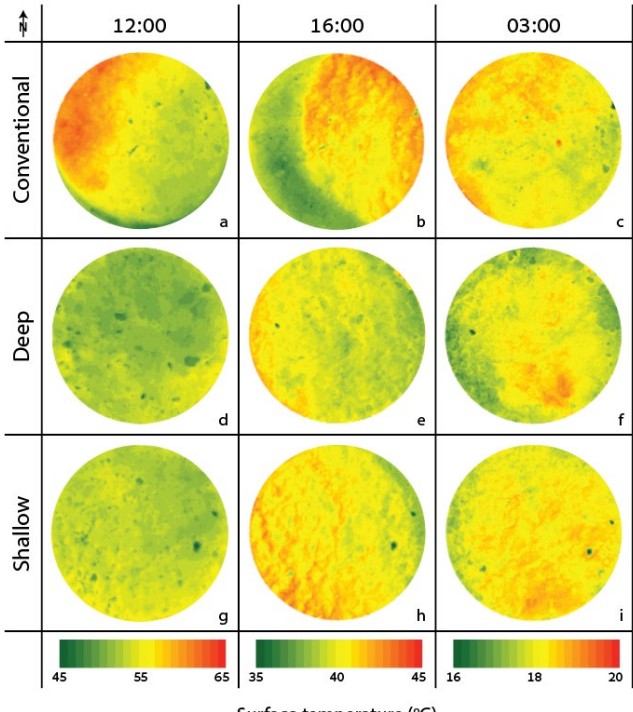

Figure 4: Thermal images of the soil surface radiometric temperature of one collar for each treatment in example hours of the day. A-C) The conventional treatment. D-F) The deep treatment. G-I) The shallow collar treatment. Note that each hour has a different temperature range.

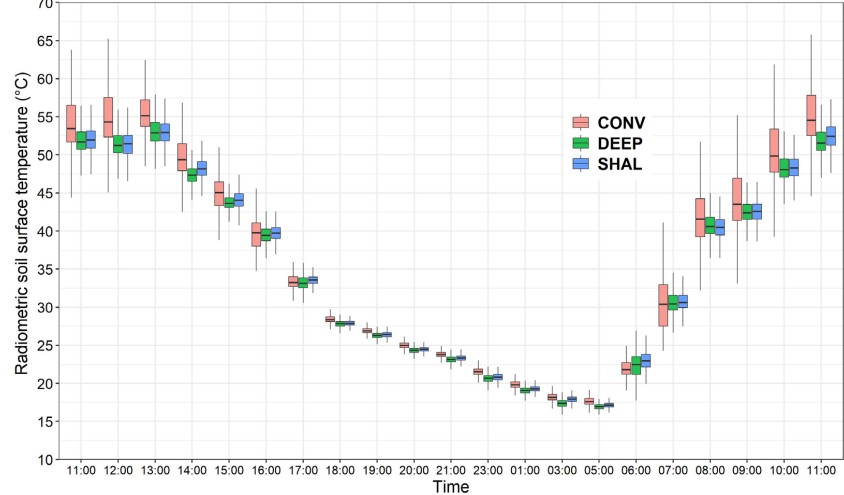

Figure 5: Box plot and whiskers of the radiometric soil surface temperatures measured within the 3 types of collars on the 17-18/08/2021.



### 3.4 The effect of the soil heat flux on soil CO₂ flux

Changes in soil surface temperature induced by the collar treatment significantly affected $F_s$. Nonetheless, $F_s$ and soil surface temperatures were uncoupled throughout the day and therefore may not be the sole variable that explains $F_s$ dynamics (Figs. 3 and 5). For example, while the soil surface temperature decreased throughout the night, $F_s$ decreased until the evening (18:00) and slowly increased during the night. However, the soil surface temperature has a prime effect on the temperature profile within the soil, as well as the direction and magnitude of soil heat flux. In fact, fig. 6 shows that $F_s$ was linearly correlated with the soil heat flux, during the night and morning efflux. Later, $F_s$ decreased earlier than the soil heat flux, resulting in a daytime hysteresis relationship (Fig. 6b).

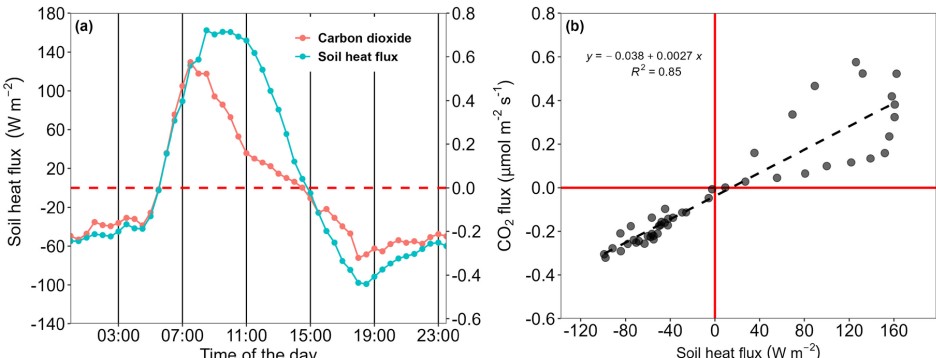

**Figure 6: Relationship between the mean days of $F_s$ and the soil heat flux for period 4 (9-16/06/2021). Note that positive $F_s$ values indicate that the direction of the flux is from the soil to the atmosphere and vice versa for negative $F_s$ values. Positive and negative soil heat flux values indicate the opposite directions than $F_s$ values.**

## 4 Discussion

Our study's results indicate that in dry and bare desert soils, using collars that protrude over the soil surface (CONV) can decrease $F_s$. This finding is consistent with a prior global assessment that identified a negative correlation between collar height above the soil surface and mean annual soil respiration rates (Jian et al., 2020). However, while we found that protruding collars resulted in significant errors of nearly 50% in $F_s$ (Fig. 3 and table 2), Jian et al. (2020) demonstrated that collar height leads to a much smaller bias of only ~10% in annual soil respiration rates. They explained this bias by nonuniform air mixing within the chamber system resulting from the larger system volume but did not consider the potential effect of elevated collars on soil surface temperatures. Moreover, 85% of the annual soil respiration rate values Jian et al. (2020) used were estimated based on a limited number of instantaneous CO₂ efflux measurements, which were usually performed during the daytime, and, therefore, overlook diurnal dynamics in $F_s$. Since $F_s$ is not constant throughout the day in desert soils but varies between daytime efflux and nocturnal uptake (Fig. 3), a small discontinuous number of daytime measurements will fail to capture errors in flux measurements. Finally, while most studies discussing potential sources of errors in $F_s$ measurements were conducted in conditions where the dominant flux is a result of microbial respiration, in dry desert soils $F_s$ is primarily driven by an abiotic process governed by changes in soil temperatures (Soper et al., 2017). Therefore, errors associated with using static chambers in dry desert soils are likely related



to alteration of geochemical processes in the soil rather than affecting the factors that influence soil microbial
activity.
The abiotic process driving nocturnal $CO_2$ uptake in dry desert soils is often explained by the combined effect of
contraction and dissolution of gaseous $CO_2$ in soil water. These processes decrease gaseous $CO_2$ concentration in
the soil surface layer, forming an atmosphere-to-soil concentration gradient and $CO_2$ diffusion into the soil (Sagi
et al., 2021; Yang et al., 2020). Contraction of soil air may decrease $CO_2$ concentration in the soil surface layer
and lead to atmosphere-to-soil pressure gradient and thermal convection, which further contributes to $CO_2$ uptake
(Ganot et al., 2014). Soil temperature negatively affects both contraction and dissolution. Higher temperature
result in less contraction and dissolution, thus a higher $CO_2$ concentration in the surface air-filled soil-pores,
ultimately leading to a smaller soil-atmosphere $CO_2$ gradient, and lower $F_s$.
The elevated walls in the CONV collars limit nocturnal radiative cooling of the topsoil layer, resulting in higher
soil temperatures that suppress the $CO_2$ concentration gradient and the actual $CO_2$ uptake from the atmosphere
(Fig. 4 and fig. 7). Following sunrise, soil temperature increases in the DEEP and SHAL collars, promoting $CO_2$
expansion and outgassing from water films, rapidly increasing $CO_2$ efflux (Fa et al., 2016). This process is delayed
in the CONV collars because the surface is entirely shaded by the collar walls (Fig. 7b), resulting in a lower mean
temperature and a narrower overall range of surface temperatures (Fig. 5; 06:00 and Fig. 7b). As a result, $CO_2$
efflux increases at a slower rate (Fig. 3). When the sun elevation increases, solar radiation is reflected off the
collar walls into the measured area, increasing the radiation flux in the unshaded soil surface and, consequently,
increasing the mean and range of soil surface temperatures compared to the DEEP and SHAL collars (Figs. 4A-
B, 5 and 7). Thus, lower surface temperatures cannot explain the significantly lower $CO_2$ efflux measured in the
CONV collars between 07:00 and 08:30. Instead, it is probably related to the significantly lower total nighttime
$CO_2$ uptake, which leads to a faster depletion of soil $CO_2$ in the following morning (Table 2).

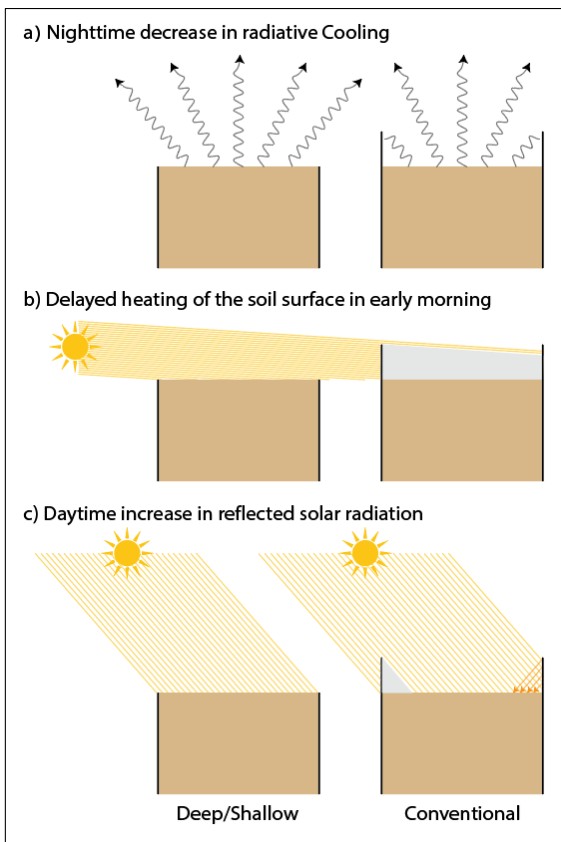

**Figure 7: Conceptual model showing the effects of collar deployment on soil surface radiative heating and cooling during the night (a), early morning (b), and daytime (c).**

The results of our study indicate that lateral diffusion is not a significant concern in dry, bare desert soils when the measurement period (i.e., the length of time during which the chamber is closed over the collar) is short, as demonstrated by the insignificant differences between $F_s$ measured over the SHAL and DEEP collars. This confirms the findings of Hutchinson and Livingston, (2001). Although statistically insignificant, the mean $CO_2$ efflux in the SHAL collars was consistently lower than in the DEEP collars between 7:00 and 14:30 (Fig. 3 and table 2). Additionally, the flux direction measured over the SHAL collars, consistently changed from efflux (positive) to uptake (negative) earlier than in the other treatments, and earlier than the soil heat flux changed from positive to negative (Fig. 6). A change in the soil heat flux sign indicates that temperatures in the uppermost soil layer are decreasing, promoting the removal of gaseous $CO_2$ from the soil air phase, followed by $CO_2$ uptake from the atmosphere. Hence, when soil temperatures are undisturbed (e.g., by the presence of a collar), we expect the onset of $CO_2$ uptake to coincide with the change in soil heat flux direction (Fig. 6). The only difference between the SHAL and DEEP collars was their insertion depth (in both the collar's top end was flashed with the soil surface). Root cutting, which is often suggested as an explanation for lower $F_s$ measured over deeper collars (Heinemeyer et al., 2011), is inapplicable when the soil is sparsely vegetated. Furthermore, our results show higher



$F_s$ values when measured over deeply inserted collars (DEEP) then when measured over shallow collars (SHAL).
Potential overestimation of $F_s$ resulting from enhanced air flow along the collar walls in the DEEP collars was
minimized by inserting the collars more than two months prior the measurements, a sufficiently long time to allow
the soil to settle around them (Hutchinson and Livingston, 2001). Lateral diffusion below the shallow collars
therefore remains the most probable explanation. As suggested by Healy et al. (1996), lateral movement likely
decreased the $CO_2$ concentration in the soil top layer during $CO_2$ efflux, decreasing the concentration gradient
between the soil and the chamber headspace, resulting in an underestimation of $F_s$. The lower soil $CO_2$
concentration beneath the SHAL collars caused the concentration gradient that drives the vertical flux to reverse
direction toward the soil, starting $CO_2$ uptake earlier than in the other treatments (fig. 3).
The conventionally deployed collars (CONV) underestimated the instantaneous $CO_2$ uptake and thus the total $CO_2$
uptake during the night (table 2). This suggests that the actual carbon sequestration by desert soils is higher than
previously reported. Theoretically, if $F_s$ in dry desert soils is derived by abiotic geochemical processes, a balanced
net daily cycle would be expected, where nocturnal $CO_2$ uptake is compensated by daytime efflux. Even in alkaline
soils, such as the ones in our study site, where the nocturnal dissolution of $CaCO_3$ may sustain $CO_2$ uptake from
the atmosphere, the reverse reaction should occur when water evaporates and $CaCO_3$ precipitates, promoting $CO_2$
efflux and system equilibrium (Roland et al., 2013). This hypothesis was supported by Hamerlynck et al. (2013)
who found that a soil in the Chihuahuan Desert, USA, only serves as a minor carbon sink (0.88 g C m$^{-2}$
accumulated over three months) and concluded that this contribution is insignificant to the global carbon balance.
Contrarily, in the Taklamakan (Yang et al., 2020) and the Gubantonggut (Xie et al., 2009) Deserts in China,
nocturnal $CO_2$ uptake led to a mean annual uptake of 7.11 and 62-622 g C m$^{-2}$, respectively. This gave rise to the
hypothesis that nocturnal $CO_2$ uptake by desert soils might explain a substantial portion of the global missing sink.
However, they did not provide a mechanism to explain where the carbon is stored, especially given that the
leaching of dissolved carbonates to groundwater is limited in space and time (Ma et al., 2014; Yang et al., 2022).
Either way, no conclusions can be drawn about the role desert soils play in the missing sink until a methodology
to measure these small fluxes is proved to be accurate. Our study shows that instantaneous $F_s$ and $F_s$ daily balance
could be significantly affected by even as small as a few centimeters difference in collar height and depth. This
implies that previous estimates of the carbon balance of desert ecosystems using static chambers need to be
carefully considered.
**5    Summary and Conclusions**
The drivers of abiotic soil $CO_2$ flux observed in dry desert soils are yet far from being understood. Further research
is needed to reconcile the discrepancy between the theoretical basis, which suggests a balanced daily cycle, and
field measurements, which often show net uptake by the soil in both diel and annual scales. Particularly, studies
should focus on improving our understanding of $CO_2$ in the soil profile in desert soils, and on allocating the
sources of water that are assumed to act as a solvent for $CO_2$ even when the soil is dry. None of these questions,
however, can be addressed without an accurate methodology to measure the small $F_s$ characterizing bare desert
soils.
During a two months measurement period in the summer of 2021, the soil in the Wadi Mashash Experimental
farm exhibited a repetitive diel cycle of $CO_2$ flux that consisted of nocturnal $CO_2$ uptake and daytime efflux,



driven by a combination of physical and geochemical processes in the soil. We show here for the first time that
collar deployment practices significantly affect this abiotic diel cycle by altering the factors that drive $F_s$. Notably,
morning $CO_2$ efflux and nocturnal $CO_2$ uptake were underestimated when measured on conventionally inserted
collars because the elevated collar walls distorted the ambient surface temperature regime. We conclude that in
bare desert soils collars should be deployed flashed with the soil surface to prevent distortion of heat exchange
between the soil and the atmosphere and between soil layers, two important drivers of the abiotic $F_s$. Lateral
diffusion under shallow collars may occur and affect $F_s$' temporal dynamics. However, we found this to be of a
lesser concern in compact soils and short measurement periods. Still, in dry desert soils, the collar insertion depth
should exceed the depth at which the fluctuations in soil $CO_2$ concentration that drive $F_s$ occur, roughly 2 cm
(Hamerlynck et al., 2013).
Deployment protocols of flux chambers should be adapted to the unique characteristics of desert soils rather than
follow standard procedures suitable for mesic environments. We conclude that using collars with at least 3 cm
length inserted flush with the soil surface will minimize measurement errors of $CO_2$ flux and will pave the way to
accurate estimates of the carbon balance of desert ecosystems.
**6     Code/data availability**
Code and data will be provided upon request.
**7     Author contributions**
**Nadav Bekin**: Conceptualization, Data curation, Formal analysis, Investigation, Methodology, Writing - original
draft. **Nurit Agam**: Conceptualization, Funding acquisition, Methodology, Project administration, Resources,
Supervision, Writing - review & editing.
**8     Competing interests**
The authors have no competing interests.
**9     Acknowledgments**
This research was supported by the Israel Science Foundation (grant number 2381/21).

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
