# Peer review of "Rethinking the deployment of static chambers for CO2 flux"

_EGUsphere, 2023_

## Author Response (AR1)

This MS presents the results of a very stong methodolgical comparison of soil collar deployment effects on aridland soil CO2 flux (Fsoil) under seasonal dry conditions. The MS is exceptionally well-written, the experimental design is straightforward and rigorous, as well as adequately tested. Studies like this that rigorously test instrumentation under field conditions instrument developers rarely account for are rare, and in my opinion, neccessary. This is especially true for aridland ecosystems, where fluxes are low, but the landscapes are vast. Getting as true an estimate of the magnitude Fsoil in these challenging environments is critical, and this study provides signal service to that goal.

We are thankful for the reviewer's positive feedback and constructive comments. Please see below the replies to the specific comments.

I only have some minor points the authors should consider in their revision:

1) Since collar geometry affects solar insolation, where on the globe Fsoil is measured matters. This study, and those of Hamerlynck et al. and Fa et. al. were made in sub-tropical to temperate zone deserts. However, there are high-latitude aridlands where abiotic uptake affects Fsoil as well (see Ball et al. 2009 Soil Biol. Biochem 41 and Parsons et al. 2004 Ecosystems 7), and low sun angles prevail. Including some discussion of undersestimating abiotic Fsoil in these ecosystems, which are faced with far greater climate change effects, is certainly worth while.

**Response**: We thank the reviewer for highlighting this point. We revised the Introduction accordingly by:

1. Removing the specific mention to warm deserts, and adding Ball el al. 2009 who worked in a cold desert to the cited papers (line 31).
2. Explicitly adding the potential effect in cold deserts (lines 56-58):

   *"These errors may intensify in high latitude cold deserts, in which the low angle of insolation will dictate a larger shaded surface area for longer periods during the day. $F_s$ was shown to be particularly affected by fluctuations in soil temperatures in cold deserts (Parsons et al., 2004; Ball et al., 2009)"*.

2) Soper et al. (2017, Geophys. Res. Letters 44) used soil column incubations and field Keeling plots to confirm the carbonate dissolution mechanisms postulated by Hamerlynck et al. 2013, so collar effects to Fsoil likely don't apply to discriminating between mechanisms. Collar effects do, as this study clear shows, have a lot to do with the actual flux magnitude.

**Response**: The reviewer is correct. The effect of the collar, through the modification of the surface temperature, indeed does not affect the mechanism, only the magnitude (in deserts, mainly the magnitude of geochemical processes). We have added an explicit mention to this important understanding in the Discussion (lines 285-286):

> *It is therefore expected that a modification of the surface temperature by the collar will affect the magnitude of the flux.*

This adds to the previous paragraph (lines 272-277), where we state that:

> *"Finally, while most studies discussing potential sources of errors in Fs measurements were conducted in conditions where the dominant flux is a result of microbial respiration, in dry desert soils Fs is primarily driven by an abiotic process governed by changes in soil temperatures (Soper et al., 2017). Therefore, errors associated with using static chambers in dry desert soils are likely related to alteration of geochemical processes in the soil rather than affecting the factors that influence soil microbial activity."*

We believe that this point is now clearer.

**3) Fa and co-workers invoked same kind of dissolution mechanisms at Hamerlynck et al., but in sandy soils (Fa et al. 2015, Hydro. Proc. 29), which showed uptake when soils were wet, not dry as in Hamerlynck et al. 2013. A brief discussion on how collar deployment might affect wet vs dry soils is warranted.**

**Response**: We see the reviewer's point. In the Introduction we present existing knowledge on the possible effects of collar height and depth on *Fs* in wet soil conditions in comparison to dry conditions (lines 61-63):

> *"Desert soils also have lower specific heat capacity than soils in humid environments due to lower water content (Hillel, 1998). The lower water content also means that a larger portion of the available energy is invested in soil heating rather than stored as latent heat during evaporation (Brutsaert, 1982)."*

Per the reviewer's suggestion, we have also added the following discussion (lines 351-355):

> *"In fact, studies show that the abiotic mechanisms involved in Fs are not restricted to dry desert conditions but rather play a significant role in Fs in deserts under wet soil conditions (Fa et al., 2016). This was found for both a semi-arid pine forest (Qubaja et al., 2020), and a temperate grassland (Plestenjak et al., 2012). Hence, the collar disruption to abiotic processes likely affects the carbon balance in various ecosystems beyond the scope of deserts during the dry season."*

**4)** Abiotic soil CO2 uptake is not limited to low-cover warm deserts and Antarctic dry valleys - see Plestenjak et al. 2012 (J Soil Sediments 12). The authors should consider taking a broader view on the global signficance of abiotic soil CO2 uptake and ecosystem carbon balance in their introduction.

**Response**: While we see the reviewer's point and agree with it, we want to emphasize that the focus of the manuscript is not the abiotic processes governing CO2 exchange between the soil and the atmosphere. Rather, we focus on the effect of the methodology on the measured fluxes. Nevertheless, we did add this point in the discussion, where it does contribute to understanding the potentially broad effect of collar insertion on measurement of Fs in various ecosystems (lines 351-355, as in the previous comment, copied here again for convinience):

> *"In fact, studies show that the abiotic mechanisms involved in Fs are not restricted to dry desert conditions but rather play a significant role in Fs in deserts under wet soil conditions (Fa et al., 2016). This was found for both a semi-arid pine forest (Qubaja et al., 2020), and a temperate grassland (Plestenjak et al., 2012). Hence, the collar disruption to abiotic processes likely affects the carbon balance in various ecosystems beyond the scope of deserts during the dry season."*

**5)** Biocrusts are also important players in arid land carbon dynamics, and their activity is measured in ways similar to soil surface flux. This is worth mentioning in the discussion section. The implications of this research goes beyond dry season Fsoil in warm deserts.

**Response**: The reviewer is correct. Indeed collar insertion may also have an effect on biologically-derived Fs. We have added this point to the manuscript 355-358):

> *"Alteration of Fs due to collar insertion is not restricted to abiotic processes. The soil biological processes, and specially activity of biological soil crust, may be significantly affected by altered soil surface conditions. Since they cover a vast area of Earth's drylands, and play a significant role in desert ecosystem's carbon balance (Wilske et al., 2008), it is important to consider these effects."*

**'Comment on egusphere-2023-714', Anonymous Referee**

The paper discusses a very important topic of correct soil respiration measurements and introduce a novel approach for improvement of measurement accuracy.

We appreciate the reviewer's support of our manuscript. Below are our responses to his constructive comments.

I do not have many notes about it. The main issue that I wanted to address is the total daily CO2 flux, which becomes negative (CO2 uptake) under deep or shallow collars installation relatively to positive flux (emission) under conventional collars installation. According to Table 2, CONV option leads to overall small CO2 emission (total efflux – total uptake) for most of periods, whereas DEEP and SHALLOW options lead to a considerable overall CO2 uptake.

**Response**: The reviewer raises an important point. It is true that numerically, the CONV resulted in a net efflux in most of the days, while the DEEP and SHAL resulted in a net uptake. It is important to note, however, that the net daily values measured by the CONV collars are very small, thus more susceptible to errors, to the point of flipping the direction. While we trust the results presented in the manuscript, we feel that concluding from the absolute daily net values must be done with caution. To clarify this point we have added the following to the manuscript (lines 328-331):

> *"In some cases, the net daily exchange measured in the CONV collars is even positive, indicating a net efflux of CO2 to the atmosphere (Table 2). Note, however, that the net daily values measured by the CONV collars are very small, thus more susceptible to errors, to the point of flipping the direction, and concluding from the absolute daily net values must be done with caution."*

In the discussion two mechanisms of soil CO2 uptake are mentioned: contraction and dissolution of gaseous CO2 in soil water (lines 275-282). The source of contraction is not mentioned. It could be trampling, but I do not think it is relevant to the desert. What else?

**Response**: The mechanism we refer to is of soil **air contraction** that occurs when the soil temperature decreases, and, following the ideal gas law, the air shrinks. We do not refer to **soil compaction** (that may occur by trampling), which leads to a decrease in total porosity. See lines 31-33 in the introduction:

> *"Researchers usually attribute this diel cycle to changes in soil temperatures and soil air pressure that leads to cycles of expansion/contraction of soil air, following the ideal gas law (Yang et al., 2020). These cycles change the surface CO2 concentration and may generate a soil-atmosphere pressure gradient (Ganot et al., 2014), both driving forces for soil CO2 flux (Fs)."*

Moreover, Qubaja et al (2020, https://doi.org/10.5194/egusphere-2023-714) showed considerable abiotic component of soil CO2 EFFLUX in the adjacent region because of CaCO3 decomposition. So, this issue must be addressed.

**Response**: We thank the reviewer for this comment. We added this point in the Discussion (lines 343-346):

> *"Furthermore, the abiotic component of Fs contributed 21% of mean CO2 efflux in a semi-arid pine forest located ~35 km north-east of our study site and therefor functioned as a source for atmospheric carbon rather than as a sink in that ecosystem (Qubaja et al., 2020)."*

**L.283: The elevated walls in the CONV collars limit nocturnal radiative cooling of the topsoil layer - Maybe rather convective than radiative, by decreasing air circulation within the collar**

**Response**: We thank the reviewer for this important note. We did consider the possibility that the elevated collar walls decrease wind speed and therefore decrease upward sensible heat fluxes. However, as shown in figure 2C, nighttime wind speed in our study site is very low (usually between 0.5-1.5 m s$^{-1}$). Hence, we concluded that air flow is a minor factor in heat exchange between the surface and the atmosphere during the night in our study site.